# Parameter-Inverted Image Pyramid Networks

**Xizhou Zhu**[2,1*], **Xue Yang**[1*], **Zhaokai Wang**[3,1*], **Hao Li**[4,1]
**Wenhan Dou**[2,5], **Junqi Ge**[2,5], **Lewei Lu**[5], **Yu Qiao**[1], **Jifeng Dai**[2,1†]

[1]OpenGVLab, Shanghai AI Laboratory   [2]Tsinghua University
[3]Shanghai Jiao Tong University   [4]The Chinese University of Hong Kong
[5]SenseTime Research

https://github.com/OpenGVLab/PIIP

## Abstract

Image pyramids are commonly used in modern computer vision tasks to obtain multi-scale features for precise understanding of images. However, image pyramids process multiple resolutions of images using the same large-scale model, which requires significant computational cost. To overcome this issue, we propose a novel network architecture known as the Parameter-Inverted Image Pyramid Networks (PIIP). Our core idea is to use models with different parameter sizes to process different resolution levels of the image pyramid, thereby balancing computational efficiency and performance. Specifically, the input to PIIP is a set of multi-scale images, where higher resolution images are processed by smaller networks. We further propose a feature interaction mechanism to allow features of different resolutions to complement each other and effectively integrate information from different spatial scales. Extensive experiments demonstrate that the PIIP achieves superior performance in tasks such as object detection, segmentation, and image classification, compared to traditional image pyramid methods and single-branch networks, while reducing computational cost. Notably, when applying our method on a large-scale vision foundation model InternViT-6B, we improve its performance by 1%-2% on detection and segmentation with only 40%-60% of the original computation. These results validate the effectiveness of the PIIP approach and provide a new technical direction for future vision computing tasks.

## 1   Introduction

In modern computer vision, high-performance image perception systems increasingly rely on large-scale pre-trained models. These models typically consume tens of thousands to millions of GPU hours during pre-training [43, 44, 41]. To adapt these expensively pre-trained models for fine-grained image perception tasks (*e.g.*, detection [4, 63, 57, 56] and segmentation [19, 50]), researchers usually combine them with image pyramids [40, 37] or feature pyramids [27, 42, 34]. This combination is crucial for constructing multi-scale features essential for image understanding.

However, integrating these pre-trained models with image pyramids results in significant computational overhead. Image pyramids process the same image at multiple resolutions with the same large-scale model, causing the computational demands to increase quadratically with the image resolutions across all scales. Although feature pyramids [27, 16, 42] aim to reduce this overhead, in MS COCO challenges [28], most top-performing models [48, 14, 64, 7] still rely on image pyramids due to their superior performance. Therefore, it is necessary to reduce the computing resources for building image pyramids while maintaining high performance.

---

*Equal contribution. †Corresponding author: Jifeng Dai <daijifeng@tsinghua.edu.cn>.

38th Conference on Neural Information Processing Systems (NeurIPS 2024).

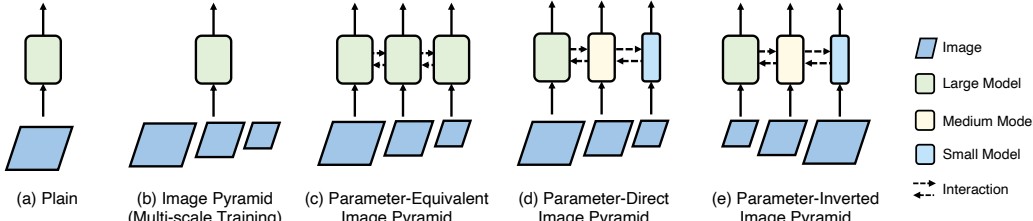

Figure 1: **Different parameter-resolution designs of image pyramid networks.** **(a)** Plain network which lacks multi-scale features. **(b)(c)** Inefficient image pyramid networks (shared weights / separate weights with interactions) using equivalently large networks for all scales. **(d)** Parameter-direct image pyramid network which processes high-resolution images with large models, leading to high computational cost. **(e)** Our efficient parameter-inverted image pyramid network (PIIP), which pairs models of increasing parameter sizes inversely with images of decreasing resolution. It delivers better performance than those of (b)(c)(d) with much lower computational cost.

To address this, our key idea is that it is unnecessary to employ vision models of equivalent size for feature extraction at all resolutions (Fig. 1(b-c)) or adopt a parameter-direct design (Fig. 1(d)). Features at different resolutions can complement each other through adequate feature fusion, thereby enhancing computational efficiency and avoiding redundant modeling of similar information. Specifically, for lower-resolution pyramid levels, the smaller images allow the efficient use of larger models to extract rich contextual and semantic features. The high-resolution branches need only provide the detail information missing from the lower-resolution features, instead of re-modeling existing semantic information. Thus, high-resolution features can focus on smaller receptive fields with less semantic information, making it possible to use smaller models to save computational resources.

Building on this strategy, a low-cost and high-performance image pyramid network can be constructed using a series of models with increasing parameter size, paired inversely with images of decreasing resolution, as shown in Fig. 1(e). Each resolution level should be able to directly leverage existing pre-trained vision foundation models for feature extraction, avoiding the large computational costs for training multi-scale image pyramid networks from scratch. In addition, sufficient feature interactions between different levels are also required to ensure the complementarity of features at different scales and avoid redundant feature extractions.

To this end, we propose Parameter-Inverted Image Pyramid Networks (PIIP) based on the complementarity of image features at different resolutions. Specifically, the network takes images at multiple scales as inputs, where higher resolution features are extracted through networks with fewer parameters for local detail perception, and lower resolution features are extracted with more parameters for global information extraction. Additionally, we introduce a feature interaction module that allows features between different resolutions to complement each other. This structure reduces the number of parameters of high-resolution branches and effectively integrates information from different receptive fields, significantly reducing computational costs without sacrificing performance.

We conduct experiments on object detection, instance segmentation, semantic segmentation and image classification. Our method achieves better performance while reducing computational costs, compared to traditional image pyramids and single-branch networks. These results validate the effectiveness of our multi-resolution feature interaction strategy and parameter-inverted paradigm and provide a new direction for future visual computing. Our contributions are as follows:

**1)** We propose a novel architecture named Parameter-Inverted Image Pyramid (PIIP) that enhances the multi-scale representational capability of vision backbones with high computation efficiency. The proposed architecture is capable of effectively and flexibly utilizing strong pre-trained vision foundation models without the need for extensive training from scratch.

**2)** We evaluate our method on classic vision tasks of object detection, instance segmentation, semantic segmentation, and image classification. Through combination of existing pre-trained models, our method surpasses single-branch models and other image pyramid methods with higher performance and lower computation cost.

**3)** To validate the generalizability of PIIP on large-scale vision foundation models, we apply PIIP to InternViT-6B [8], improving its performance on object detection and semantic segmentation by 1.9%

($55.7 \text{ AP}^{\text{b}}$) and 1.3% (59.7 mIoU) while reducing 43% and 58% of computational costs, respectively. We also provide extensive analysis and valuable insights on ablation and design guidelines for PIIP that may benefit future research.

## 2  Related Work

**Image Pyramids and Feature Pyramids.** Image pyramids and feature pyramids are two widely used techniques to enhance the multi-scale perceptive ability for downstream dense prediction tasks. Image pyramids [60, 39, 40, 37] resize the original image and extract features of different resolutions separately, allowing models to accurately detect objects of various scales. However, this technique significantly increases computational costs. Feature pyramids [27, 16, 42, 61, 34] represent another method for constructing multi-scale feature representations by merging low-resolution, semantically strong features with high-resolution, semantically weak features. Although significantly reducing computational costs, they cannot fully replace image pyramids when detecting very small or large objects [39]. Our proposed architecture integrates both image and feature pyramids and introduces the parameter-inverted paradigm to achieve efficient computation.

**Multi-branch Architectures.** Multi-branch architectures have been widely adopted to combine features from different resolutions in various computer vision tasks, including image classification [5], object detection [46, 25, 7, 52], semantic segmentation [58, 17] and multimodal dialogues [33, 21]. CrossViT [5] adopts a two-branch structure with different patch sizes to obtain inputs of various scales and different model sizes to balance the computational load. HRNet series [46, 58, 17] adopt a four-branch architecture, where the number of branches gradually increases as the layers deepen. However, they do not adopt the parameter inversion paradigm and cannot utilize existing pre-trained models. In contrast, we propose a general model architecture that supports the use of pre-trained models with different parameters to build efficient image pyramids.

**Redundancy Reduction for Visual Models.** Extensive studies focus on reducing computational redundancy for acceleration. Some work exploits the sparsity of images to accelerate model inference by reducing the number of visual tokens. Dynamic ViT [38] and AdaViT [35] design lightweight prediction modules to predict and prune less informative tokens. EViT [26] and Evo-ViT [55] compute attention scores for each token from class token to identify less informative tokens and adopt accelerated processing strategies for them. Other approaches focus on improving the model structure for efficient computation, such as attention mechanisms [47, 17, 3] or gradually reducing the spatial resolution as the number of layers increases [30, 49, 20]. Orthogonal to the above studies, we propose to use a parameter-inverted design to avoid using large models to process high-resolution images, greatly reducing the computation redundancy.

## 3  Parameter-Inverted Image Pyramid Networks

To construct efficient image pyramid networks, we employ a multi-branch structure to handle images of different resolutions with different sizes of models. As shown in Fig. 2, our architecture consists of three parts: multi-resolution branches, cross-branch interactions, and branch merging. Each branch uses an off-the-shelf pre-trained model to process images of different resolutions, where larger resolutions are processed by branches with fewer parameters. Cross-branch interactions are added every few blocks to fuse features across different feature scales. Branch merging combines the outputs from all branches to form a final output. We use the existing pre-trained ViTs [43, 44, 41] to initialize the branches, and initialize the interactions and branch merging from scratch.

### 3.1  Multi-Resolution Branches

The multi-resolution branches serve to extract representations from different image scales and semantic levels. The input image is first resized to different resolutions through bilinear interpolation, and then fed into corresponding branches to extract features at different scales. All the branches have the same number of blocks $N$, where each block contains one or multiple ViT [13] layers. Typically, blocks from different branches have different feature dimensions due to the pre-trained models, *e.g.* ViT-T, ViT-S and ViT-B. Branches with larger image sizes have a smaller number of parameters. For clarity, we refer to the branch with the largest number of parameters (with the smallest image size) as Branch 1, the second largest as Branch 2, and so on. The output of the $i$-th block of Branch $j$ is

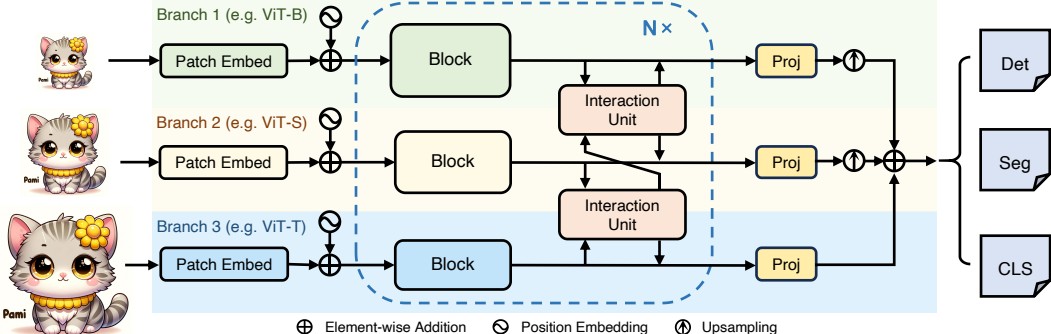

Figure 2: **Overall architecture of our method.** We use multi-resolution branches to process images of different resolutions, where larger images are handled by smaller models. Interaction Units build connections between branches. Branch merging combines the features of all branches to form the final output. Our architecture can leverage pre-trained models with different model sizes to build efficient image pyramids.

denoted as $\mathcal{F}_j^i \in \mathbb{R}^{H_j W_j / P_j^2 \times D_j}$, where $H_j$, $W_j$, $P_j$, $D_j$ are the image height, image width, patch size, and feature dimension of Branch $j$, respectively.

## 3.2 Cross-branch Interactions

Branches of different resolutions focus on different spatial scales and semantic levels. To enhance the features of different scales, we propose the cross-branch interactions. Each cross-branch interaction consists of several interaction *units*, where each unit builds connections between outputs from two feature-scale adjacent branches. The structure of the interaction unit is shown in Fig. 3.

Specifically, for the outputs of the $i$-th block of Branch 1 and 2, denoted as $\mathcal{F}_1^i \in \mathbb{R}^{H_1 W_1 / P_1^2 \times D_1}$ and $\mathcal{F}_2^i \in \mathbb{R}^{H_2 W_2 / P_2^2 \times D_2}$, we perform two deformable cross-attention [63] between the two features, denoted as $\mathrm{Attention}(\cdot)$. Each cross attention is preceded by a linear layer $\mathrm{FC}(\cdot)$ to project the feature dimension of key and value into that of the query, *i.e.* from $D_1$ to $D_2$ or vice versa. A feed-forward network $\mathrm{FFN}(\cdot)$ is added after each cross attention to provide channel-wise feature fusion. The hidden dimension ratio of FFN is set to $0.25$ to save computational overhead.

For the first cross-attention in the interaction unit, the interaction process can be formulated as:

$$\hat{\mathcal{F}}_1^i = \mathcal{F}_1^i + \gamma_1^i \mathrm{Attention}(\mathrm{norm}(\mathcal{F}_1^i), \mathrm{norm}(\mathrm{FC}(\mathcal{F}_2^i))), \tag{1}$$

$$\tilde{\mathcal{F}}_1^i = \hat{\mathcal{F}}_1^i + \tau_1^i \mathrm{FFN}(\mathrm{norm}(\hat{\mathcal{F}}_1^i)), \tag{2}$$

where $\mathrm{norm}(\cdot)$ is LayerNorm [1], $\tau_1^i$ and $\gamma_1^i$ are learnable parameters, and $\tilde{\mathcal{F}}_1^i$ is the interaction output. $\tau_1^i$ and $\gamma_1^i$ are initialized with $\mathbf{0}$ to ensure that the feature extraction of the original blocks (*i.e.* distribution of $\mathcal{F}_1^i$) will not be modified drastically due to the interactions, better utilizing the pre-trained weights.

Similarly, the second cross-attention is performed by switching the query and key/value to obtain $\tilde{\mathcal{F}}_2^i$. The outputs $\tilde{\mathcal{F}}_1^i$ and $\tilde{\mathcal{F}}_2^i$ are used for subsequent feature extractions. We only construct interaction units between each pair of feature-scale adjacent branches, such as Branch 1 & Branch 2 and Branch 2 & Branch 3.

## 3.3 Branch Merging

The final feature maps of all branches $\tilde{\mathcal{F}}_j^N$ have different spatial shapes and feature dimensions, where spatially larger feature maps have fewer feature dimensions. A single feature map fails to provide multi-scale semantic features, so we employ the branch merging module to merge the outputs of all branches into a single feature map.

As shown in Fig. 2, all branch outputs are first projected to the feature dimension of Branch 1 (the largest feature dimension) with $\mathrm{Proj}(\cdot)$. Then, all branch outputs are upsampled by bilinear interpolation $\mathrm{Upsample}(\cdot)$ into the feature map size of the last branch (the largest feature map size).

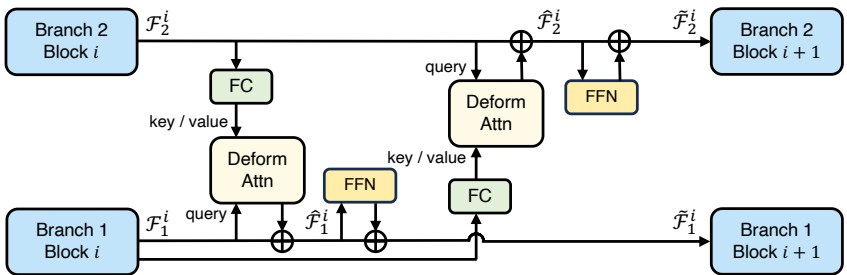

Figure 3: Structure of an interaction unit.

Finally, these outputs, with the same spatial shape and feature dimension, are added together with learnable scalar weights $w_j$ to form the final output. This process can be formulated as:

$$\tilde{\mathcal{F}}_j^{\text{out}} = \text{Upsample}(\text{Proj}(\tilde{\mathcal{F}}_j^N)), \tag{3}$$

$$\mathcal{F}^{\text{out}} = \sum_{j=1}^{M} w_j \tilde{\mathcal{F}}_j^{\text{out}}, \tag{4}$$

where $M$ is the number of branches. $\mathcal{F}^{\text{out}}$ is the final feature map, which has the largest feature resolution and also the largest feature dimension across all branches.

For object detection and semantic segmentation, $\text{Proj}(\cdot)$ is a two-convolution layer with Group-Norm [51], and the final output $\mathcal{F}^{\text{out}}$ is used for feature pyramid network [27] similar to ViTDet [23].

For image classification, we do not use the branch merging module, but instead append the original classification heads of the pre-trained models after each branch. The final classification score is the average of the output logits of all branches. We observe that using the pre-trained heads can speed up convergence compared to using a randomly initialized head after a branch merging module.

## 4 Experiments

### 4.1 Implementation Details

For comparison with Base-size models, we use pre-trained ViT-T/S/B as the branches to construct three-branch PIIP network, namely PIIP-TSB. Similarly, ViT-S/B/L are used to construct PIIP-SBL to match the computation of Large-size models. We also construct four-branch PIIP-TSBL with ViT-T/S/B/L. We set the number of interactions (each with 2 interaction units as shown in Fig. 2) $N$ to 12, *i.e.* after every layer for ViT-T/S/B or after every two layers for ViT-L. We construct multiple variants of three-branch and four-branch models with different resolution configurations. For combinations with an inconsistent number of layers, we will use a larger learning rate decay for the backbone with fewer layers. For example, for ViT-S/B (12 layers) and ViT-L (24 layers), the learning rate decay for ViT-S/B is set to be twice that of ViT-L (24/12=2).

For object detection and segmentation, we use ViT-S/B/L pre-trained on ImageNet [11] from DeiT III [44], ViT-T from DeiT [43]. ViT-H from MAE [18] and InternViT-6B [8] are used for 6B-scale experiments. For all PIIP-SBL models, we use the ImageNet-21K 384-resolution pre-trained weights to compare with previous approaches. We adopt AdamW [32] optimizer with layer-wise learning rate decay [2] to train the model on 8 NVIDIA A800 GPUs. For image classification, in Base-size experiments we use pre-trained ViT-T/S/B weights from DeiT [43]. In Large-size experiments, since DeiT does not provide ViT-L models, we use ImageNet-21K pre-trained ViT-S/B/L weights from [41].

We use the FLOPs calculation script from MMDetection [6], with our modifications to accurately calculate FLOPs of modules like self-attention and deformable attention. The script is released along with the training code. We have also manually verified the calculations using formulas, and the results are consistent with those produced by the script.

Table 1: **Comparison with baseline on COCO val2017.** We report the number of parameters and FLOPs of the backbone. Underline indicates FLOPs or metrics on par with the baseline. $AP^b$ and $AP^m$ represent box AP and mask AP, respectively.

| Model | Resolution | #Param | #FLOPs | Mask R-CNN 1× schedule | | | | | |
|---|---|---|---|---|---|---|---|---|---|
| | | | | $AP^b$ | $AP^b_{50}$ | $AP^b_{75}$ | $AP^m$ | $AP^m_{50}$ | $AP^m_{75}$ |
| ViTDet-B [23] | 1024 | 90M | 463G | 43.8 | 67.6 | 47.7 | 39.9 | 63.6 | 42.2 |
| PIIP-TSB (ours) | 1120/896/448 | 146M | 243G | 43.9 | 65.7 | 47.5 | 38.6 | 61.8 | 40.6 |
| | 1568/896/448 | 147M | 287G | 45.0 | 67.0 | 48.7 | 40.2 | 63.8 | 42.6 |
| | 1568/1120/672 | 149M | 453G | **46.6** | 68.4 | 51.1 | **41.4** | 65.2 | 44.3 |
| ViTDet-L [23] | 1024 | 308M | 1542G | 46.8 | 70.8 | 51.4 | 42.5 | 67.3 | 45.3 |
| PIIP-SBL (ours) | 1120/672/448 | 493M | 727G | 46.7 | 69.0 | 50.6 | 40.8 | 65.2 | 42.8 |
| | 1344/896/448 | 495M | 1002G | 48.2 | 71.0 | 52.8 | 42.5 | 67.3 | 45.4 |
| | 1568/896/672 | 497M | 1464G | 49.4 | 71.9 | 53.9 | 43.7 | 68.4 | 46.6 |
| PIIP-TSBL (ours) | 1344/896/672/448 | 506M | 755G | 46.9 | 69.9 | 50.6 | 41.6 | 65.9 | 44.1 |
| | 1568/1120/672/448 | 507M | 861G | 48.2 | 70.5 | 52.7 | 42.8 | 66.9 | 45.6 |
| | 1792/1568/1120/448 | 512M | 1535G | **49.6** | 72.4 | 54.2 | **44.2** | 69.2 | 47.5 |

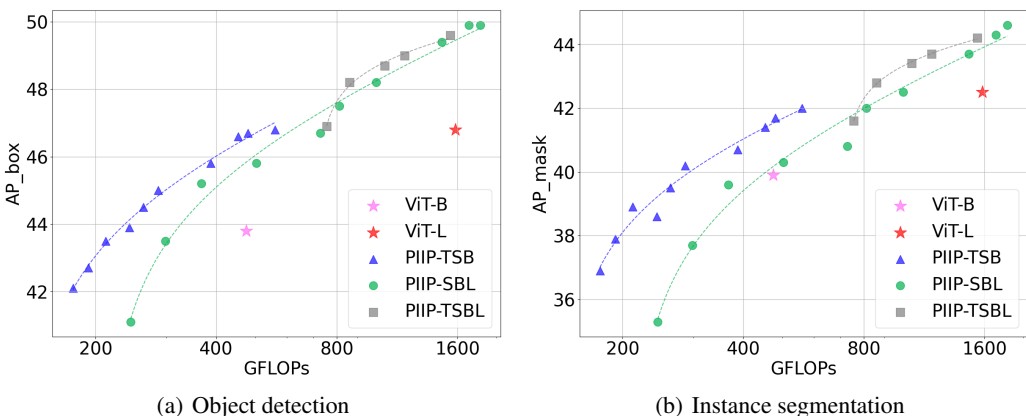

(a) Object detection  (b) Instance segmentation

Figure 4: **Performance of different PIIP variants by adjusting input resolutions.** Detailed resolution configuration and results are provided in the appendix.

## 4.2 Object Detection and Instance Segmentation

**Settings.** The MS COCO [28] dataset is used to evaluate the performance on object detection and instance segmentation. We use three detectors, including Mask R-CNN [19], Cascade R-CNN [4] and DINO [59], based on MMDetection [6]. Following common practices [7], we adopt 1× (12 epochs) or 3× (36 epochs) training schedules and use window attention [23] to save time and memory. The total batch size is 16, and the initial learning rate and weight decay are 1e-4 and 0.05.

**Effectiveness of Parameter-Inverted Image Pyramid.** To demonstrate the performance and computational advantages of the Parameter-Inverted Image Pyramid (PIIP) Networks, we perform validation on two baseline models ViTDet-B and ViTDet-L [23] in Tab. 1. Taking the three-branch structure as an example, while maintaining similar performance with ViTDet-B, our PIIP-TSB reduces the computational cost by 47.5% (243G vs. 463G) and 38.0% (287G vs. 463G) in object detection and instance segmentation tasks respectively. Similarly, compared with ViTDet-L, our PIIP-SBL reduces the computational cost by about 52.9% (727G vs. 1,542G) and 35.0% (1,002G vs. 1,542G) in the above two tasks respectively. On the other hand, with similar computational cost as the baseline, PIIP-TSB and PIIP-SBL improve the object detection performance by 2.8% and 2.6%, respectively, and instance segmentation by 1.5% and 1.2%, compared to ViTDet-B and ViTDet-L. To better illustrate the above conclusion, we depict the trend between the computational cost and performance of different PIIP model combinations by adjusting the input resolution, as shown in Fig. 4. Furthermore, when we use the four-branch structures, the curve in the figure is slightly better than that of the three-branch structure.

Table 2: **Object detection and instance segmentation performance on COCO val2017.** 'MS' means using AutoAugment [10] for multi-scale training. Large-size models use ViT weights trained on ImageNet-21K. The ViTDet-B and ViTDet-L results (and other entries) are cited from ViT-Adapter [7]. PIIP-SBL with Mask R-CNN uses higher resolutions than those in Tab. 1, as reported in Tab. 12. For PIIP-TSB with Mask R-CNN, higher resolutions (1568/896/672 -> 1792/1344/672) and a larger window size (14 -> 28) are used, compared with the results in the Tab. 1.

| Method | $AP^b$ | $AP^b_{50}$ | $AP^b_{75}$ | $AP^m$ | $AP^m_{50}$ | $AP^m_{75}$ |
|---|---|---|---|---|---|---|
| **Mask R-CNN 1× schedule** | | | | | | |
| PVTv2-B5 [49] | 47.4 | 68.6 | 51.9 | 42.5 | 65.7 | 46.0 |
| ViT-B [24] | 42.9 | 65.7 | 46.8 | 39.4 | 62.6 | 42.0 |
| ViTDet-B [23] | 43.2 | 65.8 | 46.9 | 39.2 | 62.7 | 41.4 |
| Swin-B [30] | 46.9 | - | - | 42.3 | - | - |
| ViT-Adapter-B [7] | 47.0 | 68.2 | 51.4 | 41.8 | 65.1 | 44.9 |
| PIIP-TSB (ours) | 47.9 | 70.2 | 52.5 | 42.6 | 67.2 | 45.5 |
| ViT-L [24] | 45.7 | 68.9 | 49.4 | 41.5 | 65.6 | 44.6 |
| ViTDet-L [23] | 46.2 | 69.2 | 50.3 | 41.4 | 65.8 | 44.1 |
| ViT-Adapter-L [7] | 48.7 | 70.1 | 53.2 | 43.3 | 67.0 | 46.9 |
| PIIP-SBL (ours) | 49.9 | 72.8 | 54.7 | 44.6 | 69.3 | 47.9 |
| **DINO + MS 3× schedule** | | | | | | |
| PIIP-SBL-3× (ours) | 57.9 | 76.9 | 63.3 | - | - | - |

| Method | $AP^b$ | $AP^b_{50}$ | $AP^b_{75}$ | $AP^m$ | $AP^m_{50}$ | $AP^m_{75}$ |
|---|---|---|---|---|---|---|
| **Cascade R-CNN 1× schedule** | | | | | | |
| Swin-L [30] | 51.8 | 71.0 | 56.2 | 44.9 | 68.4 | 48.9 |
| ConvNeXt-L [31] | 53.5 | 72.8 | 58.3 | 46.4 | 70.2 | 50.2 |
| PIIP-SBL (ours) | 53.6 | 73.3 | 57.9 | 46.3 | 70.3 | 50.0 |
| **Cascade R-CNN 3× + MS schedule** | | | | | | |
| Swin-B [30] | 51.9 | 70.9 | 57.0 | - | - | - |
| Shuffle-B [22] | 52.2 | 71.3 | 57.0 | - | - | - |
| ViT-B [24] | 50.1 | 69.3 | 54.3 | - | - | - |
| ViT-Adapter-B [7] | 52.1 | 70.6 | 56.5 | - | - | - |
| PIIP-TSB (ours) | 53.1 | 72.3 | 57.4 | 46.5 | 70.1 | 51.1 |
| Swin-L [30] | 53.9 | 72.4 | 58.8 | 46.7 | 70.1 | 50.8 |
| RepLKNet-31L [12] | 53.9 | 72.5 | 58.6 | 46.5 | 70.0 | 50.6 |
| ConvNeXt-L [31] | 54.8 | 73.8 | 59.8 | 47.6 | 71.3 | 51.7 |
| PIIP-SBL (ours) | 54.5 | 73.8 | 59.1 | 47.7 | 71.6 | 52.1 |

Table 3: **Experiments on the large-scale vision foundation model InternViT-6B.**

| Model | #Param | Mask R-CNN 1× schedule | | | | UperNet 160k | | |
|---|---|---|---|---|---|---|---|---|
| | | #FLOPs | Resolution | $AP^b$ | $AP^m$ | Crop Size | #FLOPs | mIoU |
| InternViT-6B [8] | 5919M | 24418G | 1024 | 53.8 | 48.1 | $512^2$ | 6105G | 58.36 |
| PIIP-LH6B (ours) | 7269M | 5643G | 1280/1024/256 | 53.5 | 47.5 | $640/512^2/192$ | 1903G | 57.82 |
| | 7271M | 10368G | 1280/1024/512 | 54.4 | 47.8 | $640/512^2/256$ | 2592G | 58.42 |
| | 7273M | 13911G | 1280/1024/640 | **55.7** | **49.0** | $640/512^2/384$ | 4560G | **59.65** |

**Results with Base-size and Large-size models.** As shown in Tab. 2, combined with Mask R-CNN, PIIP achieves higher performance than ViT-Adapter by a considerable margin, about 0.9% and 1.2% on $AP^b$. With a more powerful detector Cascade R-CNN and stronger training schedule (3× + MS), PIIP-TSB and PIIP-SBL achieve competitive performance of 53.1% and 54.5% $AP^b$, respectively. Finally, we achieve 57.9% $AP^b$ with the DINO [59] detector. These results demonstrate the scalability of PIIP.

**Results with InternViT-6B.** We further examine PIIP on an extremely large vision foundation model InternViT-6B [8]. As can be seen from Tab. 3, PIIP-LH6B finally achieves 55.7% $AP^b$ when using Mask R-CNN 1× training schedule. In addition, our PIIP can save nearly 43% of the computation and achieve better performance than the single-branch InternViT-6B by 1.9% on $AP^b$ and 0.9% on $AP^m$.

## 4.3 Semantic Segmentation

**Settings.** We use UperNet [54] as the basic framework to train on the ADE20K [62] dataset based on MMSegmentation [9]. We follow the settings of [30] to train the model for 160k iterations. The batch size, initial learning rate and weight decay are 16, 4e-5 and 0.05.

**Results with Base-size and Large-size models.** In Tab. 5, PIIP can achieve better performance with fewer computations compared with single-branch baselines. In Tab. 4, we compare PIIP with state-of-the-art segmentation backbones. PIIP-TSB attains 51.6% mIoU with UperNet, exceeding InternImage-B [48] by 1.4%. Similarly, PIIP-SBL yields 54.3% mIoU, which is outstanding compared to counterparts like ConvNeXt-XL [31] and InternImage-L [48].

**Results with InternViT-6B.** As shown in Tab. 3, similar to the conclusions obtained in the object detection experiment, our method achieves better performance than the InternViT-6B baseline with less computation. PIIP-LH6B finally achieves 59.65% mIoU without using additional optimization techniques.

Table 4: **Semantic segmentation performance on ADE20K using UperNet.**

| Method | Crop Size | mIoU |
|---|---|---|
| Swin-B [30] | $512^2$ | 48.1 |
| ConvNeXt-B [31] | $512^2$ | 49.1 |
| RepLKNet-31B [12] | $512^2$ | 49.9 |
| SLaK-B [29] | $512^2$ | 50.2 |
| InternImage-B [48] | $512^2$ | 50.2 |
| PIIP-TSB (ours) | 896/$448^2$/336 | **51.6** |
| Swin-L [30] | $640^2$ | 52.1 |
| RepLKNet-31L [12] | $640^2$ | 52.4 |
| ConvNeXt-L [31] | $640^2$ | 53.2 |
| ConvNeXt-XL [31] | $640^2$ | 53.6 |
| InternImage-L [48] | $640^2$ | 53.9 |
| PIIP-SBL (ours) | 1120/$448^2$/336 | **54.3** |

Table 5: **Comparison with baseline on ADE20K using UperNet.**

| Method | Crop Size | #FLOPS | mIoU |
|---|---|---|---|
| ViT-B | $640^2$ | 159G | 51.0 |
| PIIP-TSB (ours) | 896/$448^2$/336 | 118G | 51.6 |
| ViT-L | $640^2$ | 545G | 53.6 |
| PIIP-SBL (ours) | 1120/$448^2$/336 | 456G | 54.3 |

Table 6: **Image classification performance on ImageNet.** Underline indicates FLOPs or metrics on par with the baseline.

| Model | Resolution | #FLOPs | Top-1 Acc |
|---|---|---|---|
| DeiT-B [43] | 224 | 17.2G | 81.8 |
| PIIP-TSB (ours) | 368/192/128 | 17.4G | 82.1 |
| ViT-L [41] | 224 | 61.6G | 84.0 |
| ViT-L [41] (our impl.) | 224 | 61.6G | 85.2 |
| PIIP-SBL (ours) | 320/160/96 | 39.0G | 85.2 |
| PIIP-SBL (ours) | 384/192/128 | 61.2G | 85.9 |

Table 7: **Ablation on image pyramid and parameter-inverted design.** 'PI', 'IP' and 'Inter.' represent parameter-inverted, image pyramid and interactions. 'MS' means multi-scale training, following [10].

| Figure | Branches | PI | IP | Inter. | Resolution | #Param | #FLOPs | $AP^b$ | $AP^b_{50}$ | $AP^b_{75}$ | $AP^m$ | $AP^m_{50}$ | $AP^m_{75}$ |
|---|---|---|---|---|---|---|---|---|---|---|---|---|---|
| Fig. 1(a) | B | | | | 1024 | 90M | 463G | 43.8 | 67.6 | 47.7 | 39.9 | 63.6 | 42.2 |
| Fig. 1(b) | B | | ✓ | | MS | 90M | 463G | 44.8 | 69.2 | 49.1 | 41.0 | 65.8 | 43.9 |
| - | BBB | | ✓ | | 896/448/224 | 262M | 369G | 43.3 | 65.8 | 46.6 | 37.9 | 61.5 | 39.6 |
| - | BBB | | ✓ | | 896/672/224 | 263M | 457G | 43.8 | 66.3 | 47.3 | 38.2 | 62.2 | 39.7 |
| Fig. 1(c) | BBB | | ✓ | ✓ | 896/448/224 | 341M | 466G | 44.5 | 66.5 | 48.2 | 38.7 | 62.6 | 40.6 |
| - | TSB | | | ✓ | 896/896/896 | 148M | 468G | 44.6 | 66.4 | 48.3 | 39.0 | 62.7 | 41.4 |
| Fig. 1(d) | TSB | | ✓ | ✓ | 448/672/896 | 147M | 452G | 42.6 | 64.2 | 45.6 | 36.5 | 59.5 | 38.0 |
| Fig. 1(e) | TSB | ✓ | ✓ | ✓ | 1568/1120/672 | 149M | 453G | **46.6** | 68.4 | 51.1 | **41.4** | 65.2 | 44.3 |
| Fig. 1(a) | L | | | | 1024 | 308M | 1542G | 46.8 | 70.8 | 51.4 | 42.5 | 67.3 | 45.3 |
| Fig. 1(c) | LLL | | ✓ | ✓ | 896/448/224 | 1053M | 1458G | 46.9 | 69.7 | 51.2 | 40.8 | 65.3 | 43.3 |
| - | SBL | | | ✓ | 848/848/848 | 495M | 1539G | 47.2 | 69.4 | 51.0 | 41.1 | 65.4 | 43.7 |
| Fig. 1(e) | SBL | ✓ | ✓ | ✓ | 1568/896/672 | 497M | 1464G | **49.4** | 71.9 | 53.9 | **43.7** | 68.4 | 46.6 |

## 4.4 Image Classification

**Settings.** We load the pre-trained models for each branch and train the model for 20 epochs on ImageNet-1K [11]. The batch size, initial learning rate and weight decay are 1024, 3e-5 and 0.1. The learning rate for the random initialized interactions is 10 times the base learning rate, *i.e.* 3e-4. The other settings mainly follow the fine-tuning recipe of [44] and are provided in the appendix.

**Results.** As shown in Tab. 6, when compared with the DeiT baseline, our PIIP-SBL reduces the computational cost by 36.7% (39.0G vs. 61.6G) while maintaining the performance. When using a similar computational cost as the baseline models, PIIP-TSB and PIIP-SBL improve the top-1 accuracy by 0.3% and 0.7%, respectively.

## 4.5 Ablation Study

**Superiority of parameter-inverted image networks.** We evaluate the effectiveness of the image pyramid and parameter-inverted design by comparing our method with other methods, *e.g.* designs in Fig. 1. First of all, a single-branch with multi-scale training is the simplest image pyramid practice, as shown in Tab. 7. Compared with the baseline model, its performance improvement is limited (44.8% vs. 43.8%). Secondly, we conduct experiments by controlling the scale of the branch model and the input resolution while ensuring that the total computational cost is close. Specifically, when using the same input image resolution, the combination of models of different sizes does not bring

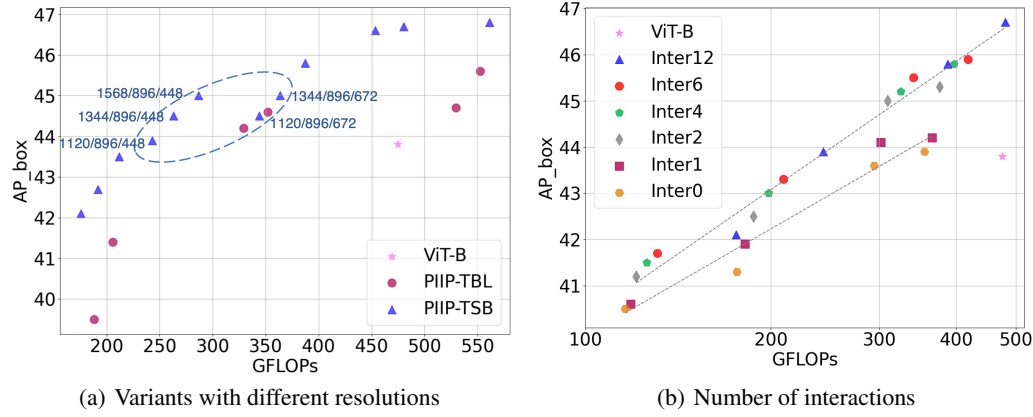

(a) Variants with different resolutions      (b) Number of interactions

Figure 5: **Ablation on model variants and number of interactions.**

Table 8: **Ablation on Branch Merging on COCO val2017.** We use PIIP-TSB 1568/896/672.

| Out Branch | $AP^b$ | $AP^m$ |
|---|---|---|
| B | 43.1 | 37.0 |
| S | 44.7 | 39.1 |
| T | 45.6 | 40.6 |
| B+S | 45.4 | 39.8 |
| B+T | 46.3 | 41.1 |
| S+T | 46.2 | 40.9 |
| **B+S+T** | **46.6** | **41.4** |

Table 9: **Ablation on attention type and number of interactions** with PIIP-TSB 1120/896/448.

| #Inter. | Regular Attention | | | | | Deformable Attention | | | | |
|---|---|---|---|---|---|---|---|---|---|---|
| | #FLOPs | $AP^b$ | $AP_l^b$ | $AP_m^b$ | $AP_s^b$ | #FLOPs | $AP^b$ | $AP_l^b$ | $AP_m^b$ | $AP_s^b$ |
| 0 | 176G | 41.3 | 59.0 | 44.6 | 22.5 | 176G | 41.3 | 59.0 | 44.6 | 22.5 |
| 1 | 211G | 41.1 | 59.1 | 44.9 | 22.6 | 182G | 41.9 | 59.8 | 45.5 | 22.4 |
| 2 | 245G | 41.7 | 59.5 | 45.2 | 22.7 | 187G | 42.5 | 60.5 | 46.4 | 23.1 |
| 4 | 315G | 41.6 | 59.2 | 45.3 | 22.8 | 198G | 43.0 | 61.0 | 47.3 | 23.3 |
| 6 | 384G | 42.1 | 59.7 | 45.8 | 23.2 | 210G | 43.3 | 61.8 | 46.9 | 23.6 |
| 12 | 592G | 42.0 | 60.0 | 45.9 | 23.1 | 243G | **43.9** | 62.4 | 47.9 | 24.4 |

significant improvements to detection performance. Correspondingly, when the three branches use the same model (*e.g.* BBB), the input image resolution is adjusted to the pyramid structure. The performance of the final model is slightly improved on $AP^b$ (44.5% vs. 43.8%), but the $AP^m$ drops significantly (38.7% vs 39.9%) due to the reduction of the maximum resolution. The former demonstrates the importance of the image pyramid, and the latter further demonstrates the need for the image pyramid to maintain a larger image scale range, which is especially essential for instance segmentation. Drawing on experience, parameter-inverted image networks are an efficient design method that can meet the above requirements, especially when compared to its opposite configuration parameter-direct image pyramid, *i.e.* TSB with 448/672/896 resolution (46.6% vs. 42.6%). As shown in Tab. 7, with less computation than the baseline, the model can support image inputs in the maximum range from 672 to 1,568, and the performance is significantly improved.

**Design guidelines for parameter-inverted image networks.** Through extensive practice, there are two empirical design guidelines when scaling up the model: 1) Prioritize increasing the image resolution of the largest image branch: as shown in the blue dashed circle in Fig. 5(a), the input resolution of the largest image branch is greatly increased without causing a sharp increase in the total computational cost. 2) The largest model does not need to exceed the compared baseline model: the introduction of larger models will limit the resolution range of the image pyramid, *e.g.* TSB is more cost-effective than TBL according to Fig. 5(a).

**Branch merging.** Experiments in Tab. 8 prove that branch merging of all branches yields the best performance by providing multi-scale semantically rich features, compared to only using feature maps from single or partial branches.

**Attention type.** The core of information interaction between branches is cross-attention. We adopt PIIP-TSB with resolution 1120/896/448 as the basic model and investigate two different attention mechanisms. As shown in Tab. 9, deformable attention [53] with linear complexity can significantly improve the performance of the model without substantially increasing the computational cost. We end up using deformable attention as the default configuration. Notably, it can be replaced by other more advanced attention mechanisms in the future to further boost performance.

Table 10: **Ablation on interaction directions** with PIIP-TSB under resolution 1120/896/448.

| Type | | | | | |
|---|---|---|---|---|---|
| #FLOPs | 210G | 230G | 230G | 243G | 283G |
| $AP^b$ | 43.5 | 43.2 | 43.6 | 43.9 | 44.0 |
| $AP^m$ | 38.7 | 38.3 | 38.6 | 38.6 | 38.7 |

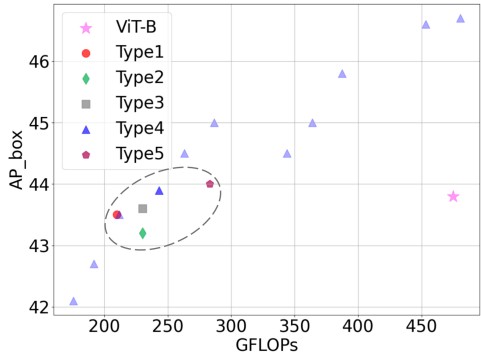

Figure 6: **Performance of different interaction directions.**

**Number of interactions.** As shown in Tab. 9, no matter which attention mechanism is used, the increase in the number of interactions will improve the performance of the model to varying degrees. Since it also increases the computational cost, we further explore the cost-effectiveness of different numbers of interactions. We conduct experiments with different resolution combinations on models with different numbers of interactions, and the scatter plot of all results is shown in Fig. 5(b). It can be seen that when the number of interactions is small (less than 2), the growth trend of model performance with the increase in computational cost is relatively slow. We attribute this to too few interactions and insufficient information complementation between branches. Therefore, we use 12 interactions by default. Note that as the model size increases (*e.g.* more layers), the number of interactions can also increase accordingly.

**Interaction direction between branches.** We compare five different interaction directions in Tab. 10. Considering both the computational cost and performance, we finally choose the fourth method, *i.e.* bidirectional connections of adjacent branches, as the default choice. As can be seen from Fig. 6, all the interaction directions achieve a satisfactory performance-computation balance, validating their ability to improve communication between branches.

## 5 Conclusion

This paper introduces the Parameter-Inverted Image Pyramid Networks (PIIP) to address the computational challenges of traditional image pyramids. With the parameter-inverted design and feature interaction mechanism, PIIP effectively balances computational efficiency and performance. Extensive experiments on detection, segmentation and classification tasks demonstrate that PIIP outperforms traditional methods and single-branch networks while reducing computational costs, providing an efficient and effective framework of multi-scale feature integration for future research.

**Limitations**. While our method manages to save computation, its memory consumption is higher than single-branch models due to the increase of parameter count. Our current method only focuses on ViT-based models. PIIP with hierarchical networks (*e.g.* CNN) or heterogeneous structures (*e.g.* CNN for some branches and ViT for other branches) remain unexplored for future work.

## Acknowledgement

This work is supported by the National Key R&D Program of China (NO. 2022ZD0161300, NO. 2022ZD0160100), by the National Natural Science Foundation of China (62376134).

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

# A Appendix

## A.1 Detailed Training Settings for Image Classification

Detailed training settings for image classification are provided in Table 11.

## A.2 Full Detection Results

Full results of Fig. 4 are provided in Table 12.

Table 11: Detailed training setting for image classification.

| | |
|---|---|
| batch size | 1024 |
| epochs | 20 |
| optimizer | AdamW |
| weight decay | 0.1 |
| learning rate scheduler | cosine |
| initial learning rate | 3e-5 |
| warmup epochs | 5 |
| mixup | 0.8 |
| cutmix | 1.0 |
| random erasing | 0 |
| auto augment | ✓ |
| color jitter | 0.3 |
| label smoothing | 0.1 |
| dropout | ✗ |
| drop path rate | 0.4 (ViT-L) / 0.2 (ViT-B) / 0.05 (ViT-S, ViT-T) |
| repeated aug | ✗ |
| gradient clip | ✗ |
| loss | cross entropy |

## A.3 Broader Impacts

Our method helps to save computational overheads of large-scale vision foundation models such as InternViT-6B, therefore reducing energy consumption. This may bring positive impacts on carbon emissions reduction and contribute to environmental sustainability. However, energy consumption of large models still needs to be treated with caution.

Table 12: Full results of PIIP variants under different resolution configurations.

| Model | Resolution | #FLOPs | Mask R-CNN 1× schedule | | | | | | | |
|---|---|---|---|---|---|---|---|---|---|---|
| | | | $AP^b$ | $AP^b_l$ | $AP^b_m$ | $AP^b_s$ | $AP^m$ | $AP^m_l$ | $AP^m_m$ | $AP^m_s$ |
| PIIP-TSB | 896/672/448 | 176G | 42.1 | 62.2 | 46.8 | 20.8 | 36.9 | 60.9 | 40.2 | 13.7 |
| | 1120/672/448 | 192G | 42.7 | 62.3 | 46.9 | 22.7 | 37.9 | 61.2 | 40.9 | 15.4 |
| | 1344/672/448 | 212G | 43.5 | 62.1 | 47.2 | 23.5 | 38.9 | 60.9 | 41.7 | 16.2 |
| | 1120/896/448 | 243G | 43.9 | 62.4 | 47.9 | 24.4 | 38.6 | 60.8 | 41.9 | 16.6 |
| | 1344/896/448 | 263G | 44.5 | 62.1 | 48.3 | 24.9 | 39.5 | 61.1 | 42.6 | 17.5 |
| | 1568/896/448 | 287G | 45.0 | 62.0 | 48.4 | 26.2 | 40.2 | 61.4 | 43.3 | 19.0 |
| | 1568/896/672 | 387G | 45.8 | 62.9 | 49.9 | 27.2 | 40.7 | 62.3 | 44.1 | 19.5 |
| | 1568/1120/672 | 453G | 46.6 | 63.1 | 50.9 | 28.5 | 41.4 | 62.3 | 45.0 | 20.6 |
| | 1792/1120/672 | 480G | 46.7 | 63.0 | 50.6 | 29.0 | 41.7 | 62.5 | 45.0 | 20.5 |
| | 1792/1344/672 | 561G | 46.8 | 62.5 | 50.8 | 30.1 | 42.0 | 62.5 | 45.1 | 21.8 |
| PIIP-SBL | 672/448/224 | 245G | 41.1 | 63.6 | 45.8 | 18.4 | 35.3 | 61.5 | 38.4 | 10.7 |
| | 896/448/224 | 298G | 43.5 | 63.9 | 47.8 | 21.9 | 37.7 | 62.4 | 41.1 | 14.3 |
| | 1120/448/224 | 367G | 45.2 | 63.7 | 49.4 | 25.2 | 39.6 | 62.9 | 42.9 | 16.7 |
| | 1120/672/224 | 504G | 45.8 | 64.7 | 50.0 | 26.1 | 40.3 | 63.3 | 43.8 | 17.4 |
| | 1120/672/448 | 727G | 46.7 | 63.0 | 50.6 | 29.0 | 40.8 | 64.4 | 44.1 | 18.1 |
| | 1344/672/448 | 811G | 47.5 | 65.8 | 51.7 | 27.6 | 42.0 | 64.7 | 45.7 | 19.5 |
| | 1344/896/448 | 1002G | 48.2 | 66.2 | 52.5 | 28.8 | 42.5 | 65.3 | 46.2 | 20.1 |
| | 1568/896/672 | 1464G | 49.4 | 66.5 | 53.9 | 30.6 | 43.7 | 64.9 | 47.5 | 22.0 |
| | 1568/1120/672 | 1709G | 49.9 | 66.9 | 54.3 | 31.7 | 44.3 | 65.3 | 48.0 | 22.9 |
| | 1792/1120/672 | 1824G | 49.9 | 65.9 | 54.3 | 32.0 | 44.6 | 65.4 | 48.3 | 23.1 |
| PIIP-TSBL | 1344/896/672/448 | 755G | 46.9 | 65.5 | 50.4 | 27.8 | 41.6 | 64.4 | 44.7 | 19.5 |
| | 1568/1120/672/448 | 861G | 48.2 | 66.1 | 52.0 | 29.4 | 42.8 | 64.7 | 46.0 | 21.0 |
| | 1568/1120/896/448 | 1052G | 48.7 | 66.4 | 52.4 | 30.2 | 43.4 | 65.2 | 46.7 | 21.4 |
| | 1792/1344/896/448 | 1180G | 49.0 | 65.9 | 52.7 | 30.5 | 43.7 | 65.0 | 47.0 | 22.4 |
| | 1792/1568/1120/448 | 1535G | 49.6 | 65.7 | 53.1 | 32.1 | 44.2 | 65.2 | 47.5 | 22.9 |

## A.4 Licenses of Datasets

**ImageNet-1k** [11] is subject to the ImageNet terms of use [45].

**COCO** [28] is subject to the Flickr terms of use [15].

**ADE20K** [62] is subject to the ADE20K terms of use [36].

