# OpenReview forum: "Parameter-Inverted Image Pyramid Networks"
_NeurIPS.cc/2024/Conference — NeurIPS 2024 spotlight_

### Official Review · Reviewer_hVKM · 2024-07-07

**Soundness:** 4
**Presentation:** 4
**Contribution:** 4
**Rating:** 9
**Confidence:** 5

**Summary:**

This paper presents a novel method named parameter-inverted image pyramid to handle the issues of high computation overhead of image pyramids. It uses different model sizes to process different resolutions. The method achieves significant results on tasks of object detection, segmentation, and classification by reducing overhead and improving performance.

**Strengths:**

1. The paper is well-organized and effectively presents its content, making for a clear and coherent read.
2. The idea of Parameter-Inverted, e.g., larger models for small images and smaller models for large images, sounds interesting.
3. The method is novel and simple without complicated handcraft structures. The direct use of existing ViT models is especially interesting and can be easily extended to other tasks.
4. The experiments are abundant and convincing. The authors provide sufficient experimental results with tables and scatter plots to verify that the proposed framework is solid and effective. The performance on 6B models is impressive.

**Weaknesses:**

1. The authors did not specify whether the FLOPs and Param in Table 1,3 include a branch merging module.
2. The discussion with other multi-resolution networks is not sufficient, e.g. HRNet [16,45,57]
3. The proposed framework can be verified on stronger pre-training, e.g. ViTDet-L (BEiTv2) vs. PIIP-SBL (BEiTv2), to further enhance the effectiveness of the method.
4. In Lines 180-181, the authors used layer-wise learning rate decay, but the authors do not indicate how to deal with ViT combinations with different numbers of layers, such as PIIP-SBL, ViT-S/B has 12 layers, and ViT-L has 24 layers.
5. The text in Figure 1 could be slightly larger.

**Questions:**

1. I am just a little curious, for PIIP-TSB， why ViT-T from DeiT but ViT-S/B from DeiT-III? Won't different pre-training methods have some impact?  According to Table 5, ViT-T with large resolution input plays the most important role, so the weaker pre-training ViT-T should limit the performance of the framework?
2. (Unimportant) In Figure 4, why are the improvements in detection better than those on instance segmentation?

---

> ### Author Rebuttal · Authors · 2024-08-06
>
> We thank the reviewer for detailed comments and suggestions and provide our response below.
> ### **W1: Whether the #FLOPs and #Param in Tables 1,3 include a branch merging module.**
> To clarify, the #FLOPs and #Param in Tables 1 and 3 contain the branch merging module, which constitutes only a small proportion (\~1%) of the entire model, similar to the #FLOPs and #Param in Table 13(a).
>
> ### **W2: Discussion with HRNet series is insufficient.**
> The HRNet series [16,45,57] employs a four-branch architecture for pose estimation, semantic segmentation, and object detection.  They also use different resolutions for each branch and add fusion layers every few blocks. However, in their architecture, the number of branches gradually increases as the layers deepen. As a result, they cannot utilize pre-trained models for different branches and must train the whole model from scratch. In contrast, our model uses a symmetric architecture for each branch, allowing for the use of pre-trained backbones. Besides, they do not adopt the parameter-inverted design that uses more parameters for processing smaller images. Lastly, the feature fusion in the HRNet series relies on convolutions and up/downsampling, which is less effective than our deformable cross-attention design.
>
> ### **W3: Verifying the method with stronger pre-training.**
> We verify our method using stronger backbones (e.g., DINOv2, BEiTv2) while maintaining the same FLOPs, and the results are consistent with those reported in the paper.
> | Backbone | Detector   | Pretrain                   | Resolution   | #FLOPs | Schedule | Box mAP | Mask mAP |
> | -------- | ---------- | -------------------------- | ------------ | ------ | ---- | ------- | -------- |
> | ViTDet-L | Mask R-CNN | BEiTv2 (L)                 | 1024         | 1542G  | 1x   | 49.3    | 44.1     |
> | PIIP-SBL | Mask R-CNN | DeiT III (S) + BEiTv2 (BL) | 1568/896/672 | 1464G  | 1x   | 51.6    | 45.4     |
> | ViTDet-L | Mask R-CNN | DINOv2 (L)                 | 1024         | 1542G  | 1x   | 46.3    | 41.7     |
> | PIIP-SBL | Mask R-CNN | DeiT III (S) + DINOv2 (BL) | 1568/896/672 | 1464G  | 1x   | 50.3    | 44.3     |
>
>
> ### **W4: Layerwise decay for different numbers of layers.**
> For combinations with an inconsistent number of layers, we will use a larger learning rate decay for the backbone with fewer layers. For example, for ViT-S/B (12 layers) and ViT-L (24 layers), the learning rate decay for ViT-S/B is set to be twice that of ViT-L (24/12=2).
>
> ### **W5: Text in Figure 1 can be larger.**
> Thank you for the suggestion. We will revise it.
>
> ### **Q1: Weaker ViT-T Pre-training limits performance.**
> We agree that initializing the small model with stronger pre-trained weights would yield better results. As we prioritize using backbones from the same family in our experiments, we sometimes face situations where a specific model size is unavailable, forcing us to use a backbone from a different family. For instance, we use ViT-S/B/L from DeiT III and ViT-T from DeiT instead because the stronger pre-trained backbone families (e.g. BEiTv2, DINOv2, DeiT III) do not provide smaller weights (e.g. ViT-T).
>
> ### **Q2: Detection is better than instance segmentation.**
> This appears to be a common phenomenon, as observed in ViT-Adapter-B/L and ViT-CoMer-B/L. We speculate that this is because the instance segmentation task is more complex than the detection task, making performance improvements more challenging, especially on a relatively high benchmark.

---

> > ### Comment · Reviewer_hVKM · 2024-08-13
> >
> > I agree with reviewer H3y6 that the parameter-inverted design is initially counter-intuitive but impressive in the end, given the experimental results. Since my concerns and questions have been addressed by the authors, and considering the high quality of this research has been recognized by all the reviewers, I keep my positive rating.

---

### Official Review · Reviewer_Lww4 · 2024-07-13

**Soundness:** 3
**Presentation:** 4
**Contribution:** 3
**Rating:** 7
**Confidence:** 4

**Summary:**

This paper proposes two techniques: 1) Models with different parameter sizes to process different resolution levels of the image pyramid. 2) A feature interaction mechanism to integrate information from different spatial scales. Extensive experiment results are used to support the claims.

**Strengths:**

Both techniques are sound. Experiments are well documented and extensive.

**Weaknesses:**

It is unclear to me if the better accuracy in table 1, 2, 3 and 4 are entirely due to the proposed techniques or it could also be partially explained by the fact that the highest resolution in the pyramid is larger. Take table 1 as an example, the input resolution for baseline is 1024. But the highest resolution for PIIP is 1792. What if you use 1792 for baseline? It is perfectly fine to first hold FLOPS constant and check accuracy, as shown in table-1. But you should also hold input constant and check accuracy.

**Questions:**

See above

**Limitations:**

yes

---

> ### Author Rebuttal · Authors · 2024-08-06
>
> We appreciate the reviewer's comments and provide additional experimental results to address these concerns.
> ### **Q1: Baseline with higher resolution is needed.**
> To evaluate the impact of using higher resolutions, we add another baseline, ViTDet-L with 1792 resolution (matching the largest resolution of PIIP-TSBL 1792/1568/1120/448), as shown in the second row of the table below. The first and third rows are from Table 1. We observed that while ViTDet-L with 1792 resolution achieves better performance compared to the 1024 resolution, its FLOPs are approximately 4 times larger. Compared with PIIP-TSBL, ViTDet-L 1792 has a lower box AP (-1.3%) and 4 times larger FLOPs. This experiment well explains that the performance improvement does not entirely come from larger image resolution. The structure of PIIP also plays an important contribution in the calculation amount and performance improvement.
>
> | Model     | Resolution         | #Param | #FLOPs | box AP |
> | --------- | ------------------ | ------ | ------ | ------ |
> | ViTDet-L  | 1024               | 308M   | 1542G  | 46.8   |
> | ViTDet-L  | 1792               | 308M   | 6458G  | 48.3   |
> | PIIP-TSBL | 1792/1568/1120/448 | 512M   | 1535G  | 49.6   |
>
> We hope that this experimental conclusion can allay your concerns and welcome further discussion.

---

> > ### Comment · Reviewer_Lww4 · 2024-08-13
> >
> > Thanks for the new results. I will keep my positive rating.

---

> > > ### Author Response · Authors · 2024-08-13
> > >
> > > Thank you for your valuable discussion and positive decision. We will make corresponding revisions in the final manuscript.

---

### Official Review · Reviewer_H3y6 · 2024-07-15

**Soundness:** 3
**Presentation:** 3
**Contribution:** 3
**Rating:** 7
**Confidence:** 4

**Summary:**

The authors propose a novel vision architecture, Parameter-Inverted Image Pyramid Networks (PIIP), which can be applied to different tasks, including classification, object detection, and instance or semantic segmentation. The authors aim to take advantage of the multi-scale information of image pyramids, without the usual excessive computational demands of processing an image in multiple different resolutions. The core idea is to use a family of networks of different computational requirements, e.g., ViT-S/B/L, and apply the lightest model (least parameters) to the image of the highest resolution in the pyramid, apply the second lighter model to the next bigger image, and so on, ending up to apply the heaviest model (most parameters) to the image of the smallest resolution. The intuition is that stronger models with more parameters can be used to extract semantically rich contextual features from images of coarser resolution, while lighter models can be efficiently applied to extract low-level features from high-resolution images. This way, the computational requirements of the different models are balanced, meaningful features from all scales are extracted, and images can be processed in higher resolution because the lighter and less expensive models are processing them.

The models are applied to the different images in the pyramid in parallel, and as features are extracted, the models communicate with each other through a proposed Interaction Unit. This way, features of different semantic meaning extracted from different models complement and inform each other. After the features from each individual model are extracted, they are merged to be used for the task at hand.

The authors perform extensive evaluation of PIIP on 4 different tasks, object detection, instance segmentation, semantic segmentation, and image classification. The authors initialize the PIIP backbones with pre-trained ViT models, and for each task they fine-tune appropriate heads. PIIP is compared to baselines of different scale, and in all experiments it demonstrates superior or comparable performance.

In addition, the authors perform multiple ablations, offering insights about the behavior of PIIP, and the importance of different design choices.

**Strengths:**

Originality:

1. The core idea of using a parameter-inverted pyramid seemed to me counter-intuitive at first, since I would expect heavier models to be applied to the images of higher resolution, which contain more information. However, the authors make a good argument about why the parameter-inverted pyramid design is sensible, and I think it is a novel idea, that in its simplicity offers a valuable contribution to the community.

Quality:

1. The authors provide extensive experiments on multiple tasks, and in their experiments they control for the computational requirements of the models, offering meaningful comparisons.

2. The experimental evaluation includes multiple ablations, which shed light on different aspects of the proposed architecture.

Clarity:

1. The manuscript is well written, and easy to follow. The authors explain the intuition and the specifics of their method in detail, accompanying the text with clear visualizations.

Significance:

1. In addition to the novelty of the proposed idea of the parameter-inverted pyramid, I think a useful conclusion of the paper is the importance of using higher resolution inputs, even if they are processed with models of smaller size. To my understanding, this is the main reason PIIP outperforms the baselines, and is highlighted by the authors in Section 4.5 (ln 260 - 266). The importance of the input resolution is not something new, but the authors offer more evidence about its impact, and importantly, they offer a new paradigm to take advantage of it.

**Weaknesses:**

Quality:

1. My main concern about this work is that the authors don’t provide actual timings and memory measurements. Computational requirements are quantified through FLOPs, however, theoretical gains in FLOPs don’t always translate to benefits in practice when the compared architectures have considerably different designs. For example, the PIIP models may be applied in parallel, but the interactions between the features may force the models to wait for each other, adding a sequential element that may add to the latency. I want to clarify that I am not claiming that this is the case, but that actual timings beyond theoretical FLOPs are needed to offer conclusive evidence, especially when efficiency is one of the main claims of this work.

2. About memory requirements, it seems to me that using a whole family of models may be prohibitive in many settings, so, memory should be explicitly measured. In addition, the authors control in their experiments for FLOPs, e.g., in Table 1 they show how PIIP models perform compared to baselines with similar FLOPs. I think it would be useful to control for memory too, e.g., in Table 1 the best PIIP models have more than 50% higher number of parameters compared to the baselines. Based on this, I think a question that naturally arises is what if the baseline was a larger model with equal parameters? If we have an extra memory budget, is PIIP the best way to use it? I would like to clarify that I don’t think the authors should necessarily have additional experiments controlling for memory, but if it is a disadvantage of PIIP, as it seems to be due to the parameter count, I think they should discuss it in the limitations.

Clarity:

1. In Table 2, the results for ViTDet-B and ViTDet-L are different compared to Table 1, why is that? Similarly, in Table 2, the results of PIIP-TSB and PIIP-SBL are different compared to Table 1. For the PIIP-SBL model, I see that the scores in Table 2 match the scores of the best model reported in Table 12, where models use higher resolutions, is this the reason for the discrepancy? However, the reported performance of the PIIP-TSB in Table 2 is higher than any performance reported for PIIP-TSB in Table 12, so, how the performance reported in Table 2 is achieved? I think the authors should clarify the configurations of the models they use in their experiments.

2. Some of the Table captions are not informative enough. For example, it is not clear why crop size is underlined in Tables 3, 6, 7. On a similar note, in the ablation about branch merging (ln 267 - 269, and Table 5), the authors don’t mention the dataset they use. I guess it is MS COCO, but I think it should be explicitly mentioned.

3. In the caption of Table 8, it is mentioned “‘PI’ and ‘IP’, ‘Inter.’ represent parameter-inverted, image pyramid and interactions”, I think it should be “‘PI’, ‘IP’, and ‘Inter.’”.

4. In Ln 262 the authors mention “green dashed circle in Fig. 5(a)”, but I think it should be “blue dashed circle”.

Significance:

1. PIIP requires a family of backbones for its processing, and the authors use pre-trained models in their experiments, while they also include some experiments on training from scratch in the Appendix. I think the need to train multiple backbones, or the need to find multiple pre-train models, may be an undesirable overhead for the adoption of the method.

**Questions:**

1. In Section 4.1 the authors provide the pre-trained architectures they used for different tasks, how did they decide which models to use?

2. What is the formula the authors use to calculate the FLOPs of the models?

3. Why there are no comparisons with state-of-the-art models on image classification?

4. In Table 8, what is the range of resolutions that the model in the second row is trained on?

**Limitations:**

The authors offer a short discussion of the limitations in Section 5, which I think should be expanded to address the memory requirements of PIIP.

---

> ### Author Rebuttal · Authors · 2024-08-06
>
> We appreciate the reviewer for providing detailed comments and highlighting our strengths. We hope our response will address the reviewer's concerns.
>
> ### **W1: Actual timing not reported; Gains in FLOPs don’t always translate to benefits.**
> We acknowledge that the reduction of FLOPs does not guarantee the improvement of throughput, which is related to engineering techniques and specific hardware. When using PIIP-LH6B with resolution 1280/1024/640, PIIP can reduce the training time of InternViT-6B from 91 hours to 62 hours while increasing the box AP from 53.8 to 55.7, though the actual increase of speed (\~32%) is less than the reduction of FLOPs (\~43%). Achieving consistent improvements across all cases can be challenging and may require additional engineering optimizations beyond the scope of this research. Previous works such as MobileNets highlighted the reduction in FLOPs as a contribution. However, the initial implementations didn't match the theoretical speed improvements. After hardware-related optimizations, current implementations have become much more efficient. We hope future work will address this challenge of PIIP.
>
> ### **W2: Control for memory; Larger baseline model with equal parameters; Discuss memory limitations.**
> Training a baseline with an equal number of parameters may be difficult. For example, there is no foundation ViT model with a parameter count equivalent to PIIP-TSB (~150M). However, we can derive insights from the 4th and 5th rows in Tab.1. PIIP-TSB obtains a similar box AP as ViTDet-L with ~50% number of parameters. Their memory consumption during training is 9.6GB and 7.9GB, respectively. We appreciate the reviewer for identifying the memory requirements as a limitation of our work and will include a discussion in the limitation section.
>
> ### **W3: Different results of ViTDet-B, ViTDet-L, PIIP-TSB and PIIP-SBL in Tables 1 and 2.**
> We thank the reviewer for highlighting the clarity issue. The ViTDet-B and ViTDet-L results (and other entries) in Table 2 are cited from the paper of ViT-Adapter, while the results in Table 1 are reproduced by ourselves.
>
> The discrepancy between PIIP-SBL results in Tables 1 and 2 is indeed from using higher resolutions, as reported in Table 12. For PIIP-TSB in Table 2, higher resolutions (1568/896/672 -> 1792/1344/672) and a larger window size (14 -> 28) are used, compared with the result in Table 1. We will include these explanations in the captions of Tables 1 and 2.
>
> ### **W4: Crop size in Tables 3, 6, 7; Dataset used in branch merging ablations.**
> We underline the crop size to ensure comparability with the baselines. The preprocessing process of semantic segmentation contains cropping from the original image, which is different from object detection that uses the whole image. In our method, the input image of Branch 2 is the cropped image, and the inputs of Branch 1 and Branch 3 are resized from the cropped image. Different initial crop sizes can lead to inconsistencies in the training data, so we annotate this to maintain consistency with the baseline settings.
> The dataset used in Table 5 and all other ablations are MS COCO. We will revise the captions for improved clarity.
>
> ### **W5, W6: Typos.**
> We shall correct them in the final version.
>
> ### **W7: Overhead of finding a family of backbones is undesirable.**
> This overhead is negligible in most cases, as the open-source community offers pre-trained models of various model sizes and families. If models of different sizes within the same family are not readily available, it is feasible to adopt models from different sources, as we demonstrated in the paper. We also conducted several additional experiments using various pre-training combinations from different sources, mainly because DINOv2 and BEiTv2 do not provide ViT-S, as shown in the table below.
> #### **Table: PIIP-SBL 1568/1120/672 using Mask R-CNN 1x schedule with different pre-trained models.**
> | Pretrain    | Box mAP | Mask mAP |
> | ------ | ----- | ----- |
> | AugReg   | 48.3    | 42.6     |
> | DeiT III   | 50.0    | 44.4     |
> | DeiT III (S) + DINOv2 (BL) | 51.0    | 44.7     |
> | DeiT III (S) + BEiTv2 (BL) | 51.8    | 45.4     |
>
> ### **Q1: How to decide the selection of different pre-trained models for different tasks.**
> In practice, we do not have strict preferences for selecting pre-trained models for different tasks. We prioritize models from the same family, and if a specific model size is unavailable, we substitute with models from other families. For instance, we use ViT-S/B/L from DeiT III, and since DeiT III does not have Tiny size model, we use ViT-T from DeiT. In the above table, we also use the AugReg pre-trained model for detection as it is used for classification in the paper.
>
> ###  **Q2: Formula for FLOPs calculation.**
> We use the FLOPs calculation script from MMDetection, with our modifications to accurately calculate FLOPs of modules like self-attention and deformable attention. We will release this script alongside the training code. We have also manually verified the calculations using formulas, and the results are consistent with those produced by the script.
>
> ###  **Q3: Comparison with SoTA on image classification is missing.**
> Due to limited time and resources, this paper does not focus on enhancing the classification performance or comparing it with SoTA. Our primary focus is on detection and segmentation tasks, which benefit more from higher input resolution and are the main target tasks of image pyramids. Besides, to maintain comparable FLOPs with the baseline, the selection space of input resolutions in classification is narrower compared to detection.
>
> ###  **Q4: Range of resolutions of "MS" in Table 8.**
> We use AutoAugment [1] for multi-scale training. Please refer to the *attached PDF* in the global response for detailed implementation.
>
> [1] Cubuk, Ekin D., et al. "Autoaugment: Learning augmentation policies from data." arXiv preprint arXiv:1805.09501 (2018).

---

> > ### Comment · Reviewer_H3y6 · 2024-08-11
> > **Thank you for your reply**
> >
> > I would like to thank the authors for their detailed response. As I mentioned in my review, my main concerns were about actual timings and memory. About timings, the numbers the authors provide are encouraging, and I agree that total agreement between FLOPs and actual runtime is not necessary from the very beginning, when a method is not yet fully optimized. I would suggest the authors include this information in Future Work or Limitations.
> > About memory, I agree that memory is not easy to control with pre-trained models, and some of the existing results allude to favorable comparisons of PIIP with baselines of similar memory. So, I find the authors' response sensible and encouraging, but in the absence of more detailed resource comparisons in the current manuscript, I will maintain my initial score.

---

> > > ### Author Response · Authors · 2024-08-13
> > >
> > > Thank you for your valuable discussion and positive decision. We will make corresponding revisions in the final manuscript.

---

### Author Rebuttal · Authors · 2024-08-06

We thank all reviewers for their time and efforts. Please refer to rebuttals under each review of detailed responses.

---

### Decision · Program_Chairs · 2024-09-25

**Decision:**

Accept (spotlight)

**Comment:**

The paper initially received 2 Accepts and 1 Very Strong Accept ratings. The rebuttal helped clarify remaining concerns and all reviewers maintained their ratings, leading to a unanimous acceptance decision. We trust that the clarifications and additional information provided in the rebuttal will be incorporated in the camera-ready version.